# GPTOpt: Towards Efficient LLM-Based Black-Box Optimization

## Abstract

Global optimization of expensive, derivative-free black-box functions demands extreme sample efficiency. Classical methods such as Bayesian Optimization (BO) can be effective, but they often require careful parameter tuning to each application domain. At the same time, Large Language Models (LLMs) have shown broad capabilities, yet state-of-the-art models remain limited in solving continuous black-box optimization tasks. We introduce **GPTOpt**, an LLM-based optimization method that equips LLMs with continuous black-box optimization capabilities. By fine-tuning large language models on extensive synthetic datasets derived from diverse BO parameterizations, **GPTOpt** leverages LLM pre-training to generalize across optimization tasks. On a variety of black-box optimization benchmarks, **GPTOpt** surpasses traditional optimizers, highlighting the capacity of LLMs for advanced numerical reasoning and introducing a flexible framework for global optimization without parameter tuning.

## 1 Introduction

Black-box optimization under tight evaluation budgets is pivotal in many scientific and engineering settings. Since derivatives are often unavailable, classical gradient-based solvers such as Newton's method or conjugate-gradient are not applicable. Practitioners therefore turn to gradient-free heuristics such as genetic algorithms, simulated annealing, and evolutionary strategies, which often require thousands of evaluations to converge (Holland, 1992; Kirkpatrick et al., 1983; Hansen & Ostermeier, 2001). When each experiment or simulation is slow or costly, such sample counts become prohibitive. Therefore, we focus on improving continuous black-box optimization under strict evaluation constraints.

Bayesian optimization (BO) (Shahriari et al., 2015) reduces this burden by fitting a probabilistic surrogate (typically a Gaussian process) and selecting queries via a heuristic acquisition function that balances exploration and exploitation. BO has driven progress in materials discovery (Erps et al., 2021), molecular design (Griffiths & Hernández-Lobato, 2020), clinical prognosis (Alaa & Schaar, 2018), and hyper-parameter tuning (Turner et al., 2021). However, BO's success hinges on hand-chosen kernels, acquisition functions, and hyperparameters whose optimal settings vary across landscapes and are difficult to tune without expert insight or extra evaluations. We show this challenge in Figure 1, where we show the gap between the performance of individual BO methods and the best of all tested BO methods averaged over multiple synthetic 5D test spaces. We see that selecting the best hyperparameters would greatly improve performance and remains a challenge of using BO in practice. Therefore, there is a need for a low-budget general-purpose global optimizer that can perform well without tuning.

The large amounts of data and increasing computing power available today have led to the development of LLMs, which provide extraordinary capabilities over a wide range of tasks. While LLMs have been used for specific optimization tasks (Yang et al., 2024; Lange et al., 2024) and assisting BO-based methods (Liu et al., 2024), their capabilities as standalone global optimizers remain limited. Experiments show that LLMs can solve basic optimization problems, Huang et al. (2024) but even state-of-the-art current LLMs remain far-behind classical methods in black-box optimization. We see these limitations in Figure 1, as we implemented an LLM-based black-box optimization scheme using Gemini 2.5 Pro (Comanici et al., 2025). Similarly to Huang et al. (2024), we find that

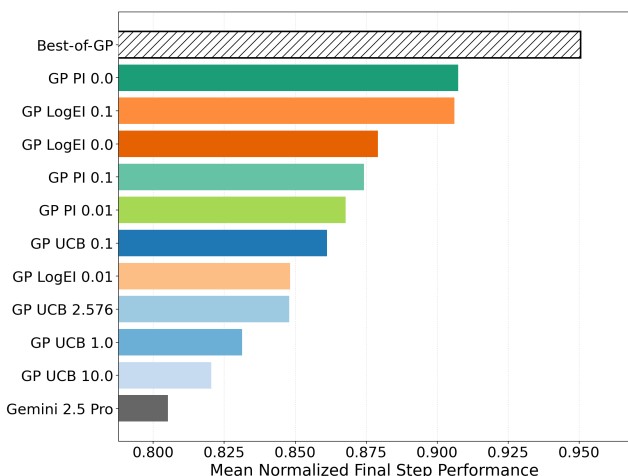

Figure 1: Performance comparison of best BO-based method to individual BO methods and SOTA LLM-based optimization over 10 5D synthetic functions. Performance is measured with a normalized regret score where higher is better.

the native LLM-based method is unable to compete with BO-based methods. We provide further details on this experiment in Appendix C.2.1.

Other works have shown that pretrained transformer-based models can used in a variety of data-driven decision making scenarios, such as machine translation (Vaswani et al., 2017), regression (Song et al., 2024b), and robotic control (Chen et al., 2021). Recent approaches have begun exploring using pretrained models for optimization, including both transformer architectures and offline pretraining (Chen et al., 2022b; Krishnamoorthy et al., 2022; Maraval et al., 2024). However, the full potential of LLMs in optimization remains untapped, in part due to the limited availability of massive high-quality optimization datasets (Song et al., 2024c). Furthermore, demonstrating robust zero-shot generalization on completely unseen problems within low evaluation budgets remains an open challenge, as most pretrained approaches require training on similar data to which the model is tested. Consequently, traditional optimization, such as Bayesian optimization, continues to be the preferred choice in practical applications. Therefore, we aim to combine the power of LLMs with high-quality optimization data to achieve performance and flexibility beyond that of current methods and add new capabilities to LLMs.

In this paper, we introduce **GPTOpt**, a LLM-based *general-purpose optimizer* for continuous black-box problems up to $10\,\mathrm{D}$. We fine-tune Llama 3.2 3B on a diverse dataset of trajectories generated using various BO variants over a synthetic function space (Dubey et al., 2024). As a result, **GPTOpt** demonstrates a robust understanding of optimization dynamics and demonstrates the ability to add black-box optimization capabilities to LLMs.

In summary, our contributions are as follows.

1. We develop a diverse *synthetic function generator* and collect *large-scale optimization trajectories* on these functions from 10 BO variants to be used as training data.

2. We design a fine-tuning approach that leverages the extensive pre-training of LLMs to add high-quality optimization capabilities.

3. Our evaluation demonstrates that **GPTOpt** achieves *strong zero-shot generalization* performance across a range of unseen global optimization benchmarks, surpassing the performance of existing state-of-the-art BO methods, while providing the opportunity for further extensions to our method.

## 2 RELATED WORK

**Bayesian optimization (BO):** BO is a powerful global optimization technique due to its ability to efficiently handle expensive black-box functions by balancing exploration and exploitation with carefully designed acquisition functions, such as Expected Improvement (EI), LogEI (Ament et al., 2023), Upper Confidence Bound (UCB) and Vizier (Golovin et al., 2017; Song et al., 2024d). Designed with different principles of trading-off exploration and exploitation, they suit different types of optimization problems. BO has been made accessible by a collection of open-source libraries and software including Spearmint (Snoek et al., 2012), BoTorch (Balandat et al., 2020), AutoOED (Tian et al., 2021), SMAC3 (Lindauer et al., 2022), HEBO (Cowen-Rivers et al., 2022), Openbox (Jiang et al., 2024), etc. However, it remains a challenge for users to determine the most appropriate configuration for their specific problem setting, as the selection is often heuristic and depends on the problem landscape.

**Pretrained optimization methods:** The idea of learning to optimize has been studied relatively early in continuous gradient-based optimization (Li & Malik, 2016; Andrychowicz et al., 2016; Chen et al., 2022a). Many early works reformulate optimization as a sequence prediction problem, training recurrent neural network (RNN) to predict the next point to evaluate. Yet, gradients need to be provided as additional information and they mostly assume in-domain settings (Chen et al., 2022a). One notable exception are Chen et al. (2017), who show that trained RNNs are able to generalize from simple objective functions to a variety of other unseen test functions without gradient information. With the increase in computation and the rise of advanced deep learning models, more recent studies are able to scale up datasets considerably and approach optimization in new ways.

Chaybouti et al. (2022) train a transformer optimizer using online meta RL, but still limited to learning task-specific solvers without generalization to a wide variety of unseen tasks. Similarly, neural acquisition process (NAP) (Maraval et al., 2024) also considers a similar online RL-based formulation of the problem. They propose an additional training objective to explicitly predicting acquisition function values to guide RL exploration more effectively. Pretrained Optimization Model (POM) (Li et al., 2024) introduces a population-based model for zero-shot optimization by formulating optimization as an evolutionary algorithm and using meta-learning for training the model, but focus on high-dimensional problems ($>$100 dimensions), which are beyond the scope we are considering.

PFNs4BO Müller et al. (2023) provides a novel black-box optimization method based on Prior Fitted Networks (PFNs), which attempts to learn a model that mimics various surrogate models. This results in a general-purpose black-box optimizer using transformers. We compare our model to PFNs4BO in Figure 3 and Figure 4. There are other works that separately study the problem of learning the surrogate (Wang et al., 2024) or the acquisition function (Volpp et al., 2019; Hsieh et al., 2021).

BONET (Krishnamoorthy et al., 2022) applies a causal transformer and RL-inspired pretraining to specific optimization tasks. They focus on synthesizing optimization trajectories by reordering samples in a given dataset. The study only evaluates domain-specific extrapolation based on a relatively large-scale dataset, instead of unseen optimization problems with just a few dozen samples.

Next, OptFormer (Chen et al., 2022b) is a text-based transformer framework designed for hyperparameter optimization for general problem space. The model is trained on a large-scale proprietary hyperparameter optimization database. The method shows promise for transformer-based models trained on large datasets by outperforming traditional solutions on the given test cases, which are drawn from similar distributions as the training data. However, the model does not start with an LLM base and therefore does not take advantage of the rich information learned from LLM pre-training. Additionally, the model shows performance degradation when tested on distributions different from its training data. However, the strong performance on a large dataset marks an important step towards general-purpose transformer-based optimization.

Lastly, RIBBO (Song et al., 2024a) is a transformer-based framework trained using offline reinforcement learning. Inspired by decision transformer, RIBBO uses conditional reward inputs to learn to optimize better than the training data provided Chen et al. (2021). Although RIBBO only evaluates results for small training and evaluation datasets from similar distributions, the ideas of conditioned transformer-based models provides an alternative step towards outperforming BO in transformer-based optimization.

**LLMs as optimizers:** Large language models have also been applied out of the box for global optimization, specifically by using in-context learning (Liu et al., 2024). These methods highlight the potential of LLMs to iteratively improve some objective, but are not specifically related to black-box optimization.

However, specifically related to black-box optimization, Huang et al. (2024) show that current LLMs are not capable of performing advanced black-box optimization. Their experiments, on a variety of state-of-the-art LLMs, show LLMs can do some basic black-box optimization tasks, but fail to come close to the performance of classical methods. In addition, Lange et al. (2024) provide a prompting strategy with the aim of making LLMs work out of the box like an evolutionary optimization method. They show their specific prompt techniques can achieve high-quality results in some limited circumstances, but primarily show results on simple function spaces and do not compare to state-of-the-art global optimization algorithms like BO.

These results show that current LLMs are not capable of state-of-the-art global optimization and while carefully designed prompting can improve results, further improvements are needed for LLMs to successfully complete optimization tasks. However, LLMs do still have a rich knowledge base and learned embeddings that we can take advantage of while teaching the LLM to optimize. **GP-TOpt** aims to provide a fine-tuning methodology that adds black-box optimization capability to LLMs, advancing both the possible use cases of LLMs and providing a new black-box optimization technique.

## 3 PROBLEM STATEMENT

For generic black-box global optimization problems, the goal is to solve $\mathbf{x}^* = \arg\min_{\mathbf{x} \in \mathcal{X}} f(\mathbf{x})$, where $\mathbf{x} \in \mathbb{R}^d$ is a vector of decision variables, $\mathcal{X} \subseteq \mathbb{R}^d$ represents the feasible search space, $f : \mathcal{X} \to \mathbb{R}$ is the black-box objective function, which is typically expensive to evaluate and lacks gradient information, and $\mathbf{x}^*$ is the global minimum of $f(\mathbf{x})$.

In practice, the optimization process begins with an initial set of points $\{\mathbf{x}_1, \mathbf{x}_2, \ldots, \mathbf{x}_m\}$ randomly sampled from the search space $\subset \mathcal{X}$, along with their corresponding objective values $\{f(\mathbf{x}_1), f(\mathbf{x}_2), \ldots, f(\mathbf{x}_m)\}$. The optimizer iteratively proposes new candidate points $\mathbf{x}_{t+1}$ based on previous points and their objective values, aiming to minimize $f(\mathbf{x})$ by exploring the search space. For example, in BO, given a dataset of points and their objective values $\mathcal{D}_t$ evaluated up to iteration $t$, $\pi(\mathcal{D}_t) = \arg\max_{\mathbf{x} \in \mathcal{X}} \alpha(\mathbf{x}; \mathcal{D}_t)$ where $\alpha$ represents a particular acquisition function. However, this process is sensitive to hyperparameter choices and choices of the GP and acquisition function.

## 4 APPROACH: **GPTOPT**

**Goal:** We aim to fine-tune a LLM that serves as a "plug-and-play" optimizer, capable of outperforming traditional black-box optimizers without hyperparameter tuning.

**Challenge:** Typically, fine-tuning a model of this nature requires a large dataset of examples (Brown et al., 2020; Bordes et al., 2024; Kim et al., 2024). Therefore, we need expert demonstrations illustrating how to choose evaluation points that minimize functions in low-budget scenarios. Unfortunately, no publicly available dataset provides such demonstrations.

**Key idea: Expert trajectories from synthetic data:** Collecting demonstrations for real-world black-box optimization is costly. Instead, we can train our model on large-scale synthetic data. By defining a design space $\mathcal{X}$ and sampling random black-box functions $f$, we generate cheap synthetic datasets. However, we still need demonstrations of how to select points to evaluate for optimization.

BO offers the ability to provide "expert" trajectories with the right parameterization. By testing many parameterizations on the same synthetic functions, we can find the best parameterizations for each individual function. We treat these best trajectories from each function space as our "expert" trajectories, which serve as the training data for our model.

## 4.1 LEARNING TO OPTIMIZE

**Optimization as sequential decision-making:** We frame black-box optimization as a sequential decision-making problem, where the optimizer acts as an agent interacting with the black-box function $f$. The optimizer observes a set of initial samples: $\{(\mathbf{x}_1, f(\mathbf{x}_1)), \ldots, (\mathbf{x}_m, f(\mathbf{x}_m))\}$, where each $\mathbf{x}_i \in \mathcal{X}$ is randomly sampled from the search space. At each step $t$, the optimizer observes all previously evaluated points and their function values, $\{\mathbf{x}_1, f(\mathbf{x}_1), \ldots, \mathbf{x}_{m+t}, f(\mathbf{x}_{m+t})\}$, selects the next evaluation point $\mathbf{x}_{m+t+1}$ based on its policy $\pi$, receives the function value $f(\mathbf{x}_{m+t+1})$, and repeats this process until the number of steps $t$ hits the user-defined limit. The agent's objective is to find a point $\mathbf{x}$ that minimizes the objective value $f(\mathbf{x})$ over its interactions with the environment.

**Synthetic function generators:** Due to the lack of available real-world test functions, we aim to create realistic and diverse synthetic functions to generate our training data. Therefore, we use 5 different classes of synthetic functions, while also adding additional augmentations on top of the original functions. By creating functions with these diverse methods, we provide a base for the model to learn realistic function dynamics across a variety of possible function spaces.

We generate synthetic functions using five methods, each designed to capture different kinds of structure. Gaussian processes provide smooth landscapes with controllable correlations. Random neural networks create complex nonlinear surfaces shaped by layered transformations. Ordinary differential equations simulate dynamical systems to produce rich temporal dynamics. Expression trees combine symbolic operators to form interpretable but varied functions. Fourier expressions use sums of sinusoidal components to generate oscillatory, multi-scale patterns. On top of these bases, we apply augmentations such as nonlinear warps, discontinuities, and periodic ripples to increase complexity, ensuring the model trains on a wide spectrum of challenging function behaviors.

**Trajectory generation:** For each of the synthetic functions generated ranging from 2D to 10D, we run BO with 10 acquisition variants. These include LogEI (Ament et al., 2023), Upper Confidence Bound (UCB), Probability of Improvement (PI) acquisition functions with various exploitation-exploration parameters. For LogEI and PI, we adapt the exploitation-exploration level with a range of 3 parameters and for UCB we use a range of 4 parameters. The goal is to provide a wide range of state-of-the-art methods that work across a diverse range of functions. Further details on the BO implementation is available in Appendix C.2.2. Each model is initialized with 10 random points before iteratively fitting the GP and selecting new points using the acquisition function. We generate 10 trajectories per environment, each consisting of 10 initial samples followed by 40 optimization steps, for a total of 50 evaluations.

We generate trajectories on functions ranging from 2D to 10D, with most of the data coming from 2D to 6D due to computational limits. We generate 5,000 synthetic functions for each function subclass, totaling 50,000 functions per dimension for 2D to 6D. Due to computational limitations, we generate 100 functions per subclass, totaling 1,000 functions per dimension for 7D to 10D. Given we generate 10 trajectories per function, this totals around 2.5 million total trajectories.

## 4.2 MODEL ARCHITECTURE

We use the Llama 3.2 family of models as our base model (Dubey et al., 2024). Specifically, we use the 3B model for our experiments. This is a text-only LLM that provides a model with reasonable capabilities, but is manageable to fine-tune within our compute budget. We fine-tune the model using low-rank adaption (LoRA) with Unsloth Hu et al. (2022); Daniel Han & team (2023). This provides a fast and efficient framework for fine-tuning LLMs, which allows us to train models within our computational limits.

### 4.2.1 TOKENIZATION

After generating training data using BO, we convert these trajectories into a textual format for the LLM. To do this, we take advantage of the extensive LLM pre-training and utilize the native numerical tokens. The Llama 3.2 models have numerical tokens between 0 and 999, before which numbers are split into multiple tokens. We find that small models like Llama 3.2 struggle to complete simple math with numbers represented by multiple tokens. Therefore, to minimize the number of tokens used and to reduce complexity, we convert all number values in the trajectory to integers in the range of 0 to 999, meaning each value uses one token. This takes advantage of the rich pre-training and

complex numerical representations that the LLM has learned for each number. We begin with a few descriptors about the problem, including some parameter setup (number of random steps, model steps). The actions are scaled linearly within their possible range before being discretized into integers. For each trajectory, the highest objective value is given a score of 999 and the lowest a score of 0, with all other values linearly discretized. Lastly, for each action-objective pair, we include a True-False value indicating whether the trajectory achieves its best value up to that point or not.

Therefore, each step is represented as

$$[a_0, a_1, \ldots, a_{d-1}] : s, \ \mathbf{1}\{\text{new best value achieved}\}.$$

We show sample of an example prompt below.

---

**Example Prompt**

### Instruction:
This problem is a synthetic 2D black-box optimization problem. We will begin by initializing with 10 random steps, after which you must optimize the objective with 20 additional steps. Random Steps: Step 1:[765,488]:210,True; Step 2:[192,128]:251,False; Step 3:[136,651]:611,False; Step 4:[350,526]:220,False; Step 5:[370,666]:226,False; Step 6:[160,924]:999,False; Step 7:[20,451]:760,False; Step 8:[576,854]:209,True; Step 9:[686,983]:227,False; Step 10:[667,414]:207,True.
### Response:
Optimization Steps: Step 1:[422,581]:208,False; Step 2:[684,642]:211,False; Step 3:[257,276]:235,False; Step 4:[446,640]:206,True; Step 5:[738,266]:269,False; Step 6:[462,736]:207,False; Step 7:[440,616]:206,False; Step 8:[449,682]:207,False; . . .

---

### 4.2.2 TRAINING

Our fine-tuning methodology treats learning both action and objective values as a sequential modeling problem. We represent the "expert" trajectories in text form and fine-tune the Llama 3.2 model using Unsloth. From the full set of trajectories, we select the $k$ best trajectories at step counts of 5, 10, 15, 20, 25, 30, 35, and 40 for each function. The best trajectories are determined by the minimum value achieved within the given number of steps. After converting these top-$k$ trajectories into the text format, the dataset consists of $8 \times k \times 250{,}000$ trajectories, where 8 corresponds to the number of step counts, $k$ is the number of selected trajectories, and 250,000 is the number of synthetic functions. We fine-tune the Llama 3.2 3B model with a batch size of 128 for one epoch. We evaluate multiple values of $k$ (see Section 5.4) and find that $k = 5$ yields the best results. Additional implementation details and hyperparameters are provided in Appendix A.

### 4.2.3 INFERENCE

At inference time, we use the same prompt schema as at train time, except for the objective normalization. Because we do not have access to the range of objective values our method will achieve we have to determine a scaling strategy for the objective values. We also utilize an acquisition function given multiple forward passes of the model to select the action with the best predicted chance of providing an improved point. The combination of these two methodologies allows our model to outperform the best individual BO methods.

**Objective Normalization:** Our model represents objective values as discrete integers in the range $[0, 999]$. During training, each trajectory is normalized using its observed minimum and maximum values. At inference, however, the true global minimum and maximum are unknown, so we approximate the scaling adaptively. Specifically, we set the maximum value of 999 to the largest value observed so far in the trajectory, and define the minimum as $C_{\min}(t)$, where $t$ is the current step. We initialize $C_{\min}(0) = 500$ and decrease it linearly to $C_{\min}(T) = 100$ for the final step $T$. All other values are linearly interpolated in this range. This schedule mimics the decreasing range observed during training without relying on the unknown true optimum.

**Acquisition Function:** To further improve model performance, we take advantage of the model's predicted objective value distribution. We run $k$ forward passes on the model to generate $k$ possible actions and $k$ predicted objective value distributions. We find that $k = 4$ is a good balance of

performance relative to runtime cost. We then use an expected improvement acquisition function relative to the previously achieved minimum to select the proposed action with the highest expected improvement. This allows the model to select best predicted point from its options.

Therefore to complete inference, we begin by randomly sampling points and converting the trajectory history to our prompt format. We then run multiple forward passes of the model to get multiple possible actions, using the acquisition function to select a single action. We evaluate the selected action and provide the updated information to the model, iteratively running the model, selecting a sample point, and evaluating until the budget is exhausted.

## 5  EXPERIMENTS

### 5.1  BENCHMARKS AND EXPERIMENTAL SETUP

We test our methodology on in-distribution and out-of-distribution functions. We utilize our function generators from model training to create a holdout test set of diverse functions to test in-distribution performance. To test out-of-distribution performance, we use the Virtual Library of Simulated Experiments (VLSE) (Surjanovic & Bingham, 2013) and Black-Box Optimization Benchmark (BBOB) (Elhara et al., 2019). These both contain functions that are traditionally used as standard benchmarks for global black-box optimization. We test our method on 2D to 10D function spaces.

We utilize the same global optimizers used to generate training data as our baselines. This includes the following acquisition functions: LogEI, UCB, and PI with various parameterizations. These baselines were implemented in BoTorch (Balandat et al., 2020). We also compare to other gradient-free optimizers, including Covariance matrix adaptation evolution strategy (CMA-ES), particle swarm optimization (PSO), tree-structured Parzen estimator (TPE), differential evolution (DE), and PFNs4BO, a general purpose transformer-based optimization framework (Müller et al., 2023).

We show the performance of each method on these benchmarks using mean normalized performance. Normalized performance evaluates the performance of each method in relation to the overall regret while factoring in the scale of the function. We measure this with $P = (\min(\{f(\mathbf{x}_1), \ldots, f(\mathbf{x}_L)\}) - f^*)/(f_m - f^*)$, where $f^*$ is the global minimum and $f_m$ is the median of the initial randomly sampled points. We report the mean performance over tested functions.

### 5.2  PERFORMANCE ON HOLDOUT TRAINING DISTRIBUTION

We show the performance of the model on holdout training distribution functions in Figure 2. This includes functions from each training class in dimension 2D to 10D, Gaussian process, neural network, ODE, expression tree, and Fourier expression, along with augmentations for each class. We see that **GPTOpt** far outpaces any individual classical optimization method.

### 5.3  PERFORMANCE ON OUT-OF-DISTRIBUTION BENCHMARKS

We show the performance of the model on the BBOB and VLSE test suites over dimensions 2D to 10D in Figure 3 and Figure 4. We see that **GPTOpt** outperforms each individual BO method over 40 steps for both test suites. In addition, other than CMA-ES performing well over initial steps on VLSE functions, the performance of **GPTOpt** is superior to each BO method across the step progression, showing robust optimization performance over steps. This is an important result because **GPTOpt** was not trained on any BBOB or VLSE functions and this is a true out-of-distribution test.

### 5.4  ABLATIONS

**Top-k Trajectories:** We evaluate the effect of filtering trajectories using different top-$k$ values during training data selection, where $k$ represents the top-$k$ trajectories selected from each training function. Specifically, we train models with $k \in \{1, 2, 5\}$, each for one epoch. Due to computational constraints, we were unable to train a model on the complete dataset with $k = 10$. Results are shown in Figure 5a, where we observe that using more data improves performance.

**Inference Method:** We next ablate different inference methods. Our model has several parameters that affect performance, most notably the sampling parameters. One key factor is the temperature

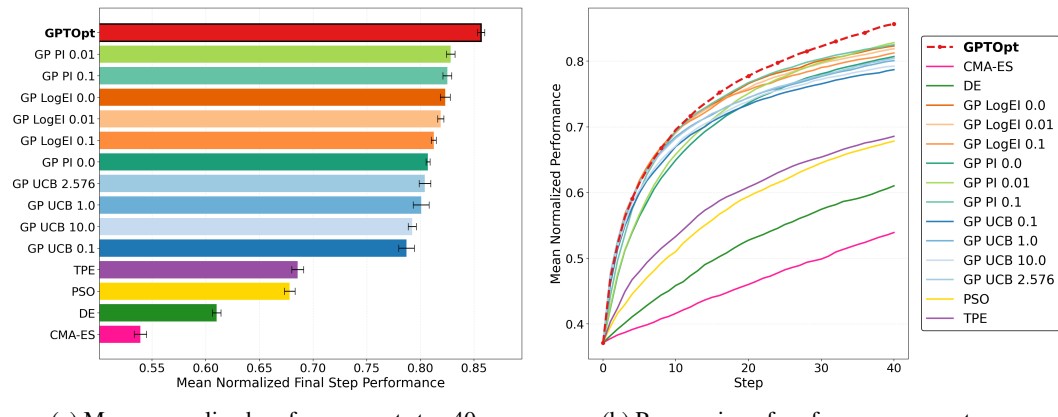

(a) Mean normalized performance at step 40.

(b) Progression of performance over steps.

Figure 2: Mean normalized performance with standard error over 5 splits on holdout training distribution test functions from 2D to 10D. We test over 10 functions of each type from each dimension, totaling 900 overall functions.

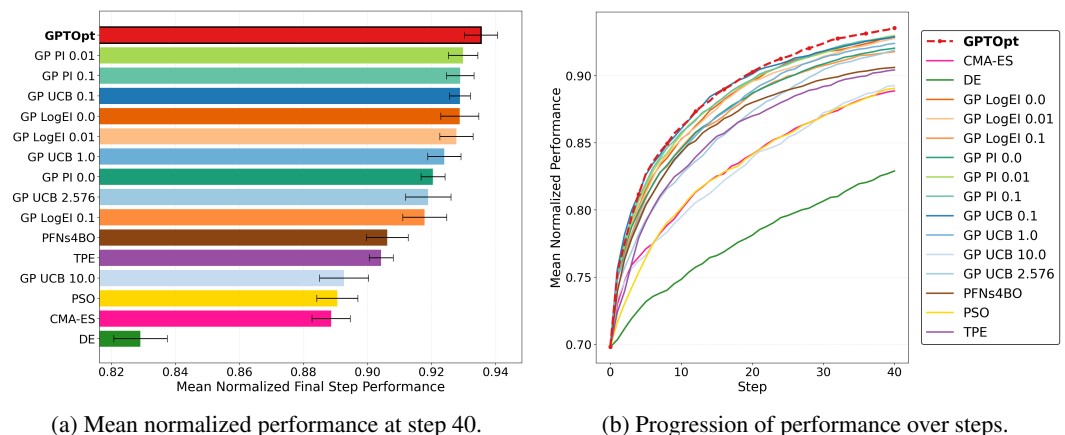

(a) Mean normalized performance at step 40.

(b) Progression of performance over steps.

Figure 3: Mean normalized performance with standard error over 5 splits on out-of-distribution BBOB 2D to 10D test functions. We test over 50 functions of each type from each dimension, totaling 450 overall functions.

of the sampling distribution, which controls the level of randomness in the generated points. We test temperatures $t \in \{1.0, 1.5, 2.0\}$ and find that $t = 1.5$ yields the best performance. Results are shown in Figure 5b.

## 6 CONCLUSION, LIMITATIONS, AND FUTURE WORK

In this work, we introduced **GPTOpt**, a novel approach for black-box global optimization using a fine-tuned LLM. We formulated the optimization task as a sequential decision-making problem and created a wide variety of synthetic functions for training. Learning from high-quality trajectories from multiple expert optimizers, our model shows strong performance on unseen optimization problems without the need for task-specific tuning. Therefore, we show a methodology that adds optimization capabilities to LLMs, taking advantage of their rich pre-training stage to improve upon previous learned methods. This work highlights the potential of LLMs in advancing global optimization and naturally paves the way for promising future extensions.

Our approach is currently limited to continuous, single-objective optimization less than 10D, which restricts its applicability to problems involving combinatorial or mixed-integer decision spaces, as well as multi-objective scenarios. Extending the model to tackle these broader categories could

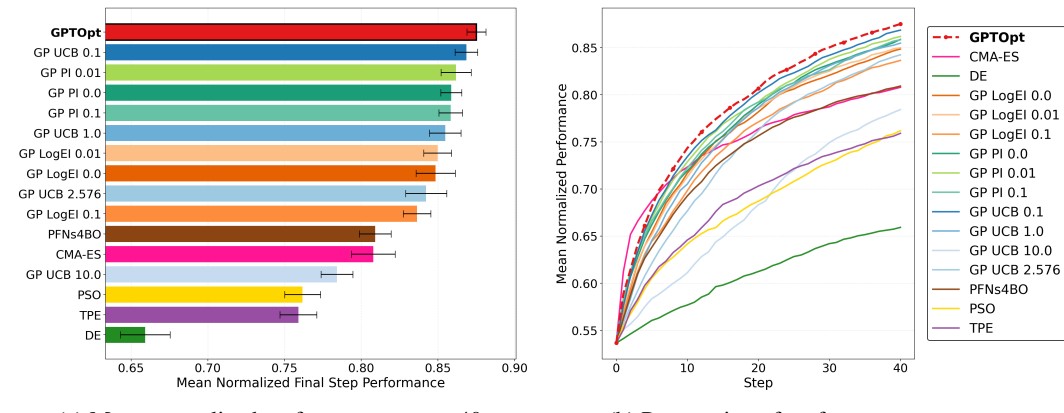

(a) Mean normalized performance at step 40.

(b) Progression of performance over steps.

Figure 4: Mean normalized performance with standard error over 5 splits on out-of-distribution VLSE 2D to 10D test functions. We test over 50 functions of each type from each dimension, totaling 450 overall functions.

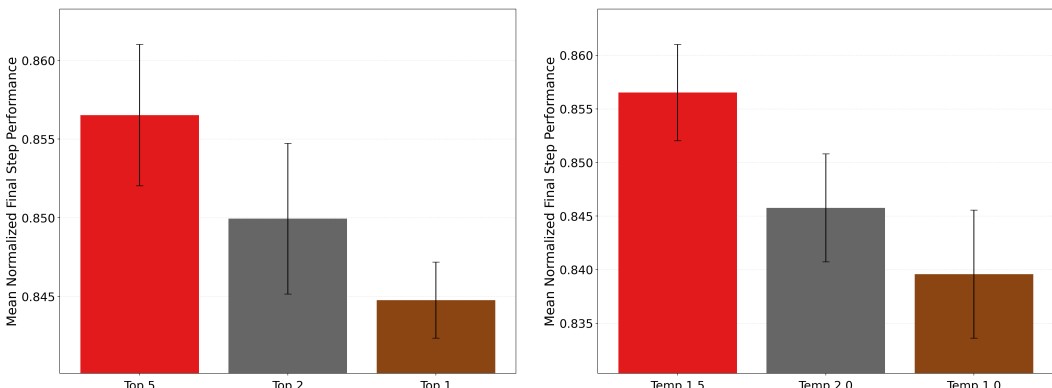

(a) Ablation on top-k trajectories. We show the mean normalized optimization performance with standard error over 5 splits of 3 top-k models tested after 40 steps on 2D to 10D holdout training distribution functions.

(b) Ablation on sampling temperature. We show the mean normalized optimization performance with standard error over 5 splits of 3 temperature parameters tested after 40 steps on 2D to 10D holdout training distribution functions.

Figure 5: Ablation studies on training dataset and inference method.

significantly enhance its versatility. Finally, although **GPTOpt** beats all baseline BO variants over traditional benchmarks, extending to other distributions that have more complex relationships or behavior may degrade performance. Adding more diverse training data or fine-tuning further to specific domains may further improve generalizability and performance.

However, there are many advantages of our LLM-based approach in comparison to BO and opportunities for future work. First, if training data is available, **GPTOpt** could be finetuned to meet the needs of individual circumstances. This could result in further improved performance, even with limited training data. Additionally, **GPTOpt** could be augmented to include semantic information that is difficult to incorporate with BO-based methods. Including information such as historical data from previous similar tests or parameter names and definitions could provide valuable information that models based on **GPTOpt** could take advantage of that BO or other classical methods could not. Lastly, scaling up with additional training data from a broader range of functions, as well as utilizing larger models may improve performance. Therefore, **GPTOpt** provides a base that can be utilized to extend to different contexts or quantities of information in a way that BO cannot. Making these extensions possible is a valuable part of our contributions.

## 7 ETHICS STATEMENT

We do not believe this work has significant ethical concerns and is solely related to improving the performance of black-box optimization methods. We utilized LLMs in this work as a formatting aid.

## 8 REPRODUCIBILITY STATEMENT

We provide details on the implementation and use of our model, both in training and inference, throughout the paper and with further details in Appendix A. Additionally, we will provide the code, fine-tuned model, and dataset as open-source upon acceptance of this paper.

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

# A  IMPLEMENTATION DETAILS

## A.1  DATA GENERATION

To construct a diverse training dataset, we implement multiple classes of continuous synthetic black-box functions within a unified environment interface. Each environment is initialized with a dimensionality and a random seed for reproducibility. Inputs are normalized to the continuous range $[-1, 1]^d$, enabling all generators to share the same interface for sampling, evaluation, and visualization. For reproducibility, all parameters and random draws are controlled by fixed seeds.

**Gaussian Processes (GPs)**  GP-based functions are sampled from a Gaussian process prior defined over the input space. Random covariance kernels are generated by combining base kernels (RBF, Matern, Rational Quadratic, Exponential) using addition and multiplication, with up to three kernels per function. Hyperparameters such as lengthscales and variances are drawn from log-uniform distributions. A fixed set of initial points is sampled, and function values are generated from the corresponding multivariate Gaussian. Evaluations are obtained by conditioning on this prior, yielding smooth but diverse optimization landscapes.

**Random Neural Networks (NNs)**  NN-based functions are defined by randomly initialized fully-connected feedforward networks with 5–10 hidden layers and 16–256 hidden units per layer. Activation functions are drawn at random from a pool including ReLU, Tanh, and LeakyReLU. No training is performed; instead, the output values are fully determined by initialization, producing highly nonlinear but stable functions.

**Ordinary Differential Equations (ODEs)**  ODE-based functions are generated by simulating small dynamical systems of dimension 2–6. The system evolves according to

$$\frac{dy}{dt} = A(z)y + \tanh(B(z)y) + U(z) + f_{\text{forc}}(t, z),$$

where the coefficients depend linearly on the input $z$. For each evaluation, the system is integrated forward in time using a fourth-order Runge–Kutta scheme over 100–200 steps. The final state is projected through a random readout vector to produce a scalar objective value, introducing temporal dependencies and nonlinearities tied to the input.

**Expression Trees**  Expression-tree functions are constructed by combining elementary operations (addition, multiplication, sine, tanh, and polynomial terms) into randomly generated symbolic formulas. Each tree includes a normalized linear base term to prevent trivial constants, plus a number of nonlinear and polynomial features. Random input rotations and optional output warps (ex: tanh) are applied to further increase variety. This produces interpretable symbolic expressions with highly diverse behaviors.

**Fourier Expressions**    Fourier-based functions are generated by summing 5–50 sinusoidal terms of the form

$$f(x) = \sum_i A_i \sin(\omega_i^\top x + \phi_i),$$

where amplitudes $A_i$, frequencies $\omega_i$, and phases $\phi_i$ are sampled uniformly from predefined ranges. This results in oscillatory functions with varying frequencies and interference patterns, creating landscapes with multiple local optima.

**Augmentations.**    To further enrich the base function classes, we apply a probabilistic augmentation layer that introduces nonstationary and real-world-like behaviors. Augmentations include:

- **Nonlinear input warps:** smooth affine transformations with $\tanh$ nonlinearities.
- **Discontinuity staircases:** injecting sharp sigmoidal jumps at random thresholds.
- **Nonlinear kinks:** adding softplus hinges and odd-power bends to create local irregularities.
- **Soft plateaus:** differentiable quantization that snaps values toward discrete centroids.
- **Frequency-modulated ripples:** superimposing oscillatory signals with varying frequency and amplitude.

Each augmentation is applied with a fixed probability and scaled relative to the natural range of the base function, ensuring meaningful variation without overwhelming the underlying structure.

We use a variety of global optimization algorithms in each environment to generate expert trajectories. For each of the synthetic functions generated ranging from 2 D to 10 D, we run BO with 10 acquisition variants. These include LogEI (Ament et al., 2023), Upper Confidence Bound (UCB), Probability of Improvement (PI) acquisition functions with various exploitation-exploration parameters. For LogEI and PI, we adapt the exploitation-exploration level with a range of 3 parameters and for UCB we use a range of 4 parameters. For LogEI and PI, we use $\xi = [0.0, 0.01, 0.1]$ and for UCB we use $\kappa = [0.1, 1.0, 2.576, 10.0]$, which are all standard values for these acquisition functions. We begin by sampling 10 initial points randomly to initialize each model. We then fit the GP model to the set of points and use the specified acquisition function to sample a prospective point. After evaluating that point, we refit the model and continue iteratively. We generate trajectories on functions ranging from 2D to 10D, with most of the data coming from 2D to 6D due to computational limits. We generate 5,000 synthetic functions for each function subclass, totaling 50,000 functions per dimension for 2D to 6D. We then generate 100 functions per subclass, totaling 1,000 functions per dimension for 7D to 10D. Given we generate 10 trajectories per function, this totals around 2,500,000 total 40 step trajectories.

We generate trajectories using CPU machines, primarily on an Intel Xeon Platinum 8260 system. We find that, although BoTorch supports GPU acceleration, the most cost-efficient manner to generate trajectories is to use parallel CPU processes. However, even with optimizations, running our baseline global optimizers is slow, especially for higher dimensions. Therefore, we total $\sim 50,000$ vCPU hours for our data generation. We were limited by the amount of trajectories we could generate and more data may help improve the model.

### A.2    TRAINING

Our full training dataset contains $\sim 2,500,000$ total trajectories. We use data augmentation to expand this and enable further generalization. We augment by swapping axes, flipping the action space, and randomizing the order of the initial points to provide additional training data. This augmentation greatly expands the trajectory space, particularly for higher dimensions. We also include shortened trajectories (i.e., the first 20 steps of a 40 step trajectory) to expand the dataset. Overall, Table 1 contains the number of environments and therefore trajectories of each type before augmentation. We input trajectories of length 5, 10, 15, 20, 25, 30, 35, and 40 to the model, where length does not include the 10 random initial steps.

We use the Llama 3.2 family of models as our base model (Dubey et al., 2024). Specifically, we use the 3B model for our experiments. This is a text-only LLM that provides a model with reasonable capabilities, but is manageable to fine-tune within our compute budget. Expanding the size

Table 1: Number of functions and trajectories for each dimension.

| Dimension | Functions | Trajectories |
|---|---|---|
| 2D–6D | 50,000 | 500,000 |
| 7D–10D | 1000 | 10,000 |
| **Total** | **254,000** | **2,540,000** |

of the base model is a possibility for further improvement. We fine-tune the model using low-rank adaption (LORA) with Unsloth Hu et al. (2022); Daniel Han & team (2023). This provides a fast and efficient framework for fine-tuning LLMs, which allows us to train models within our computational limits. We use a custom data collator to convert our trajectory data into text-based format with augmentations.

We use the hyperparameters shown in Table 2 to fine-tune the model. The model is trained using 4 Nvidia H100 GPUs. In total, training takes $\sim 1.5$ days on this system.

Table 2: Hyperparameters for model training.

| Hyperparameter | Value |
|---|---|
| Base Model | Llama 3.2 3B |
| Learning Rate | $2 \times 10^{-4}$ with cosine scaling |
| Weight Decay | 0.01 |
| Batch Size | 128 trajectories |
| Lora R | 64 |
| Lora Alpha | 64 |
| Bias | None |
| Dropout | 0.0 |
| Precision | BF16 |
| Epochs | 1 |

## A.3 INFERENCE

At inference time, we use the same prompt schema as at train time, except for the state normalization. Because we do not have access to the range of objective values our method will achieve we have to determine a scaling strategy for the objective values. We also utilize an acquisition function given multiple forward passes of the model to select the action with the best predicted chance of providing a improved point. The combination of these two methodologies allows our model to outperform the best individual BO methods.

**Objective Normalization:** Our model represents objective values as discrete integers in the range $[0, 999]$. During training, each trajectory is normalized using its observed minimum and maximum values. At inference, however, the true global minimum and maximum are unknown, so we approximate the scaling adaptively. Specifically, we set the maximum value of 999 to the largest value observed so far in the trajectory, and define the minimum as $C_{\min}(t)$, where $t$ is the current step. We initialize $C_{\min}(0) = 500$ and decrease it linearly to $C_{\min}(T) = 100$ for the final step $T$. All other values are linearly interpolated in this range. This schedule mimics the decreasing range observed during training without relying on the unknown true optimum. This is important because we encourage the model to further improve upon the previous states reached.

**Acquisition Function:** To further improve model performance, we take advantage of the model's predicted objective value distribution. We run $k$ forward passes on the model to generate $k$ possible actions and $k$ predicted objective value distributions. We find that $k = 4$ is a good balance of performance relative to runtime cost. We then use an expected improvement acquisition function to select the proposed action with the highest expected improvement. This uses the predicted probability distribution of each proposed action and calculates the expected value of the improvement relative to the previously achieved minimum. This allows the model to select better actions than any individual BO method. This is because we use the model's learned knowledge of possible actions with promising regions to explore to improve upon its training data.

Using these methods, we convert the model's history to our prompt format. We then sample multiple possible actions, using the acquisition to select a single action. We evaluate the selected action and provide the updated information to the model, iteratively selecting points until the budget is exhausted.

## B  FURTHER EXPERIMENTS

We provide further experiments to demonstrate the optimization abilities of **GPTOpt**.

### B.1  DIMENSION BREAKDOWN

We provide a breakdown of the performance of **GPTOpt** over 2D to 10D for both out holdout test functions and the out-of-distribution benchmarks in Figure 6. We see that **GPTOpt** is consistently the top performing method, with some variability due to randomness. This provides further evidence of the robustness of our method, as **GPTOpt** is consistent across all dimensions tested.

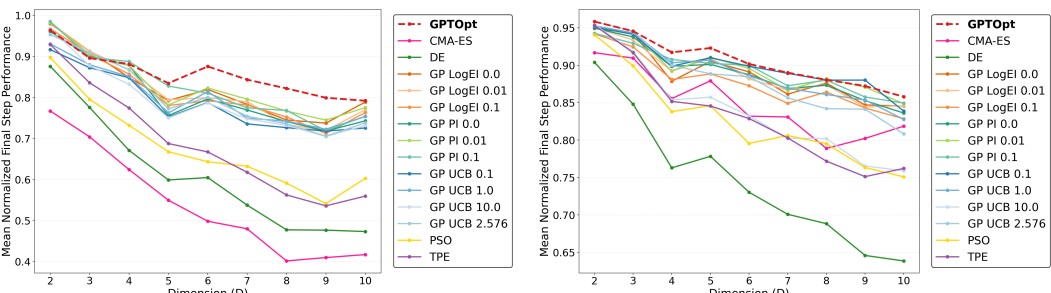

(a) Dimension progression over holdout training distribution test functions per dimension.

(b) Dimension progression over BBOB and VLSE test functions per dimension.

Figure 6: Mean normalized performance after 40 steps on test functions split over dimensions.

### B.2  WIN RATE

We show the win rate of **GPTOpt** against all baselines on our test suites in Figure 7. We see that **GPTOpt** outperforms all baselines in win-rate. This is another important metric for an optimizer and highlights the robustness of **GPTOpt**.

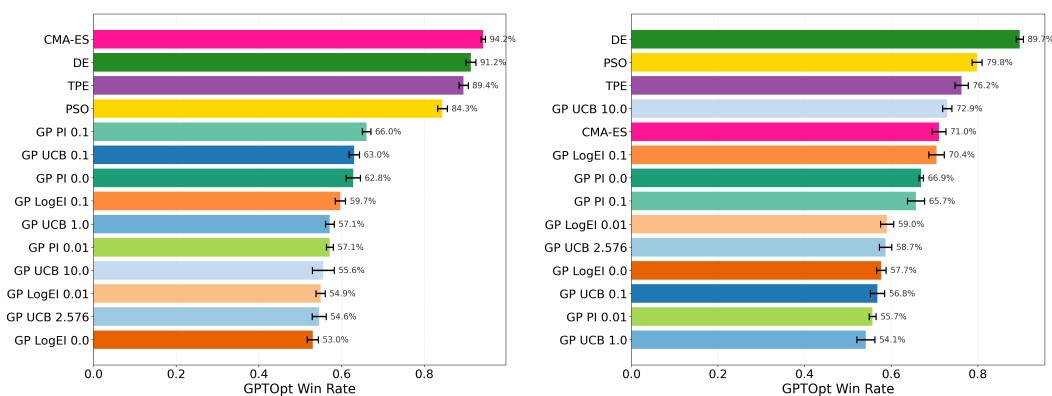

(a) Win rate with standard error of **GPTOpt** compared to all baselines after 40 steps on 2D to 10D holdout training distribution test functions.

(b) Win rate with standard error of **GPTOpt** compared to all baselines after 40 steps on 2D to 10D BBOB and VLSE training distribution test functions.

Figure 7: Win rate after 40 steps of **GPTOpt** compared to all baselines.

## C  EVALUATIONS

We provide further detail into the evaluation strategy used for both benchmarks and baselines in our experiments.

### C.1  BENCHMARKS

**GP:** We use the function generator used for generating training data as the initial baseline for our method. This contains diverse GP functions over a flexible dimension.

**BBOB (Elhara et al., 2019):** The Black-Box Optimization Benchmarking suite was developed as part of the COCO (Comparing Continuous Optimizers) platform to provide a rigorous and standardized environment for evaluating continuous, unconstrained optimization algorithms. BBOB includes 24 benchmark functions that represent a wide range of challenges encountered in real-world optimization, such as separability, multimodality, ill-conditioning, and non-convexity. Each function is parameterized with randomized shifts, scalings, and rotations to prevent algorithms from overfitting to specific patterns. The suite was carefully designed through mathematical constructions and transformations of base functions to create controlled yet diverse test cases. It supports varying dimensions and is widely used in the black-box optimization community.

**VLSE (Surjanovic & Bingham, 2013):** The Virtual Library of Simulation Experiments is a benchmark suite aimed at simulating real-world optimization problems where the objective function is defined by computational simulations rather than closed-form expressions. We use the optimization test problems from this set.

### C.2  BASELINES

We aim to provide state-of-the-art baselines for the various benchmarks on which our model is tested.

#### C.2.1  LLM EVALUATION

Similar to Huang et al. (2024), we explore the use of state-of-the-art LLMs as out-of-the-box black-box optimizers. We explore various possible prompt formats with Gemini 2.5 Pro (Comanici et al., 2025). We find that this model is not capable of matching the performance of BO or other classical black-box optimization methods, but is capable of some optimization performance. We use the following prompt format for Gemini, which provides the model with contextual information about the problem and the evaluated points so far. The model responds with a specific format that corresponds with our expected inputs.

---

**Example Prompt**

You are an expert AI optimization assistant.
Your objective is to find the input 'x' that **minimizes** the output of a black-box function $f(x)$.
The input parameters and their bounds are:
- 'param_0': a float between $-1.0$ and $1.0$
- 'param_1': a float between $-1.0$ and $1.0$
Here are the points evaluated so far:
- Point 1: 'param_0': 0.532, 'param_1': -0.023 $\rightarrow$ 22.801
- Point 2: 'param_0': -0.616, 'param_1': -0.743 $\rightarrow$ 3902.753
- Point 3: 'param_0': -0.728, 'param_1': 0.303 $\rightarrow$ 37473.902
- Point 4: 'param_0': -0.3, 'param_1': 0.053 $\rightarrow$ 1015.296
- Point 5: 'param_0': -0.26, 'param_1': 0.333 $\rightarrow$ 1581.922
- Point 6: 'param_0': -0.68, 'param_1': 0.849 $\rightarrow$ 73664.680
- Point 7: 'param_0': -0.959, 'param_1': -0.097 $\rightarrow$ 51350.996
- Point 8: 'param_0': 0.152, 'param_1': 0.711 $\rightarrow$ -46.642
- Point 9: 'param_0': 0.373, 'param_1': 0.967 $\rightarrow$ 1647.743
- Point 10: 'param_0': 0.335, 'param_1': -0.171 $\rightarrow$ -205.972

---

> - Point 11: 'param_0': 0.271, 'param_1': -0.219 → -232.627
> - Point 12: 'param_0': 0.207, 'param_1': -0.267 → -241.180
> - Point 13: 'param_0': 0.143, 'param_1': -0.315 → -233.962
> So far, the best point found is 'param_0': 0.207, 'param_1': -0.267 with a value of -241.180.
> Based on all this information, suggest the next point to evaluate.
> Your response MUST be **only a Python dictionary** mapping parameter names to values.
> Example: 'param_0': 0.1, 'param_1': 0.1

We iteratively provide this prompt format with the updated evaluations until the evaluation budget in exhausted. We find that the model provides reasonable suggestions but remains significantly worse than the performance of the BO methods.

### C.2.2 BO EVALUATION

We utilize the same global optimizers used to generate training data as our baselines. This includes the following acquisition functions for BO, implemented in BoTorch (Balandat et al., 2020):

- **Log Expected Improvement (LogEI)**: A variant of EI that operates in log space, making it more suitable for objectives with large dynamic ranges or multiplicative noise.
  Parameters: $\xi = [0.0, 0.01, 0.1]$

- **Probability of Improvement (PI)**: Chooses points that maximize the probability of achieving an improvement over the current best observation, often leading to more exploitative behavior.
  Parameters: $\xi = [0.0, 0.01, 0.1]$

- **Upper Confidence Bound (UCB)**: Prioritizes points with high predicted mean and uncertainty, controlled by a trade-off parameter.
  Parameters: $\kappa = [0.1, 1.0, 2.576, 10.0]$

### C.2.3 OTHER OPTIMIZERS

We also compare to other gradient-free global optimizers. We use implementations from Akiba et al. (2019) where available.

- **Covariance Matrix Adaptation Evolution Strategy (CMA-ES)**: An evolutionary algorithm that adapts the sampling distribution using covariance information for efficient search.

- **Particle Swarm Optimization (PSO)**: A population-based stochastic optimizer inspired by the social behavior of birds and fish, adjusting candidate solutions based on personal and global bests.

- **Differential Evolution (DE)**: A population-based method that perturbs candidate solutions using scaled differences between population members.

- **Tree-structured Parzen Estimator (TPE)**: A sequential model-based optimization method that builds probabilistic models of good and bad configurations and samples promising candidates by maximizing expected improvement.

- **PFN4BO Müller et al. (2023)** : A prior-fitted network method that learns the surrogate of various possible surrogate models. We use their code and pre-trained HEBO plus model from their repository in our implementation.

These classical methods typically require thousands of iterations to converge but provide a strong point of comparison to highlight the performance of our model.

