# OpenReview forum: "GPTOpt: Towards Efficient LLM-based Black-Box Optimization"
_ICLR.cc/2026/Conference — Submitted to ICLR 2026_

### Official Review · Reviewer_p1Ys · 2025-10-19

**Soundness:** 2
**Presentation:** 2
**Contribution:** 1
**Rating:** 2
**Confidence:** 5

**Summary:**

The paper introduces GPTOpt, a method for black-box optimization that involves fine-tuning a 3B parameter language model (Llama 3.2) on a large, synthetically generated dataset. The training data consists of optimization trajectories from various Bayesian Optimization (BO) configurations, where the "best" performing trajectory for each synthetic function is used as an expert demonstration. The goal is to create a "plug-and-play" optimizer that requires no hyperparameter tuning. The authors evaluate GPTOpt on holdout synthetic functions as well as standard out-of-distribution benchmarks (BBOB, VLSE), demonstrating that it outperforms individual BO methods and other learned optimizers in low-dimensional settings (up to 10D).

**Strengths:**

- The paper tackles the well-known challenge of hyperparameter sensitivity in BO
- The goal of creating a robust, tuning-free optimizer is valuable
- On the selected low-dimensional benchmarks (BBOB, VLSE), GPTOpt demonstrates impressive performance

**Weaknesses:**

- The conceptual framework is not original, as the idea of "learning to optimize" with transformers is already well-established and several recent works have explored the LLM+BO integration. For example, OptFormer [1] already established the paradigm of treating BBO as a text-based sequence modeling problem and this work follows this template directly.
-  The authors mentioned that they mapped a continuous space to 1000 discrete points per dimension, which in my eyes makes the method fundamentally incapable of finding precise optima. This design choice makes the optimizer impractical for any real-world application where precision matters
- The method is only demonstrated on problems with up to 10 dimensions, which is relatively low-dimensional. Many important optimization problems in science and engineering actually involve much higher-dimensional spaces

**Questions:**

- Could the authors comment on why other prominent learned optimizers and LLM-based BBO frameworks, such as OptFormer and LLAMBO, were not included as baselines in the experiments?
- How does GPTOpt handle problems where the true optimum lies between the discrete points of your grid?
- What is the primary bottleneck that limits the method to <10D? Is it context length, the cost of data generation, or a fundamental issue with the model's ability to learn in higher dimensions?
- Given that the model is trained on trajectories generated by BO, is it possible that a simple ensemble of the baseline BO methods (e.g., selecting the best proposal from all 10 variants at each step) could achieve comparable performance?

---

> ### Author Response · Authors · 2025-11-21
>
> Thank you for your time in reviewing our submission and for the constructive feedback you provided.
>
> **Motivation**
>
> We wish to clarify our motivation and justify our design choices in GPTOpt. The core motivation of using LLMs in black-box optimization is to make advanced black-box optimization more flexible and easier to use. While Bayesian optimization (BO) performs well in many scenarios, challenges with implementation, selection of hyperparameters, and lack of explainability prevent many real-world users from using these methods. Instead, users turn to design of experiments (DoE) approaches, which are inefficient and expend valuable resources that could be saved if more advanced techniques were used [1, 2]. The challenges with implementing BO into workflows is seen when users select poor parameters or integrations with their experimental design. Other works attempt to simplify access to BO methods, stating that “libraries typically aren’t oriented towards experimental research, resulting in limited crossover between tutorial content and experimentalist needs”, showing the gap between theoretical results and usability in practice [3]. This gap between expert use and practical use suggests there is a need for a high-quality black-box optimization methodology that is more flexible for real-world scenarios and can be more widely utilized in practice [4, 5]. This motivation is further explored in a position paper by Song et al. [5], which “advocates for wider research and adoption of Transformers and LLMs for black-box optimization.” These efforts emphasize the idea that BO is difficult to use in practice and the community would benefit from a more flexible method.
>
> LLMs are very flexible with various types of inputs and can integrate well with many users’ workflows. However, as mentioned by many reviewers, they generally struggle with precise numerical reasoning. With GPTOpt, we develop a methodology that can overcome these challenges and produce high-quality optimization trajectories using a flexible, text-based input scheme. We envision this as one step in the path towards truly flexible and easy to use black-box optimization, as using an LLM provides the potential for a multitude of extensions. These extensions could include providing additional information about an experiment a user is performing, such as relevant papers or documentation, previous experiments that are similar, or multi-modal information about a process. Additionally, pretrained models bring structural priors, natural language reasoning, and task generalization that purely scratch-trained models do not capture. With this in mind, GPTOpt demonstrates that end-to-end LLM-based optimization is not only feasible but highly competitive.
>
> **Limitations and Ablations**
>
> While our methodology is currently limited to continuous black-box optimization with 10 or fewer dimensions, this is not a fundamental limitation of our approach and is instead a limitation of our computation. Most black-box problems that can be realistically optimized within 50 evaluations are within our scope. However, our methodology is flexible to dimension and could extend to higher dimensional scenarios with more compute. Additionally, expanding to discrete or categorical optimization problems would be a possible extension within our framework. While these extensions were not possible within our scope and compute limitations, the flexible framework of GPTOpt allows for these modifications to be possible.
>
> Regarding ablations, our scope was similarly restricted by compute limitations. We aimed to justify as many of our design decisions as possible, but retraining our model for each design feature was not possible within our limitations. We have tested our methodology on a larger model, Llama 3.1 8B, in Table 1. We find that this improved results beyond the performance of our 3B parameter model, suggesting that further scaling or expansion to a reasoning model may result in further improved performance.
>
> Table 1:
> | Model | 2D - 10D BBOB and VLSE Mean Normalized Performance |
> |-|-|
> | Llama 3.2 3B | 0.8921 ± 0.004 |
> | Llama 3.1 8B | 0.8974 ± 0.004 |
>
> [1] Pickles, Thomas, et al. "Comparative study on adaptive Bayesian optimization for batch cooling crystallization for slow and fast kinetic regimes." Crystal Growth & Design 24.3 (2024): 1245-1253.
>
> [2] Rummukainen, Hannu, et al. "Traditional or adaptive design of experiments? A pilot-scale comparison on wood delignification." Heliyon 10.2 (2024).
>
> [3] Baird, Sterling G., Andrew R. Falkowski, and Taylor D. Sparks. "Honegumi: An Interface for Accelerating the Adoption of Bayesian Optimization in the Experimental Sciences." arXiv preprint arXiv:2502.06815 (2025).
>
> [4] Liu, Tennison, et al. "Large language models to enhance bayesian optimization." arXiv preprint arXiv:2402.03921 (2024).
>
> [5] Song, Xingyou, et al. "Position: Leverage foundational models for black-box optimization." arXiv preprint arXiv:2405.03547 (2024).

---

> > ### Author Response · Authors · 2025-11-21
> >
> > We also wish to respond to your specific questions, along with the updates to our motivation and framing above.
> >
> > **Questions**
> >
> > Q1: We agree that including comparisons to OptFormer and LLAMBO would provide helpful insight and comparison to the performance of GPTOpt. We have previously attempted to test OptFormer with our codebase, but found their pretrained model ran far too slow to complete our tests without extensive access to TPU machines. We have begun integrating and testing LLAMBO as a baseline and will include comparisons in the revision. However, the primary goal of GPTOpt is to show that LLM fine-tuning is a possible and promising route for further improvements in black-box optimization. We also show that state-of-the-art LLMs are not capable of this high-quality optimization without fine-tuning. Lastly, Bayesian optimization remains the state-of-the-art in general purpose black-box optimization and we compare our model with a diverse range of BO parameterizations that encompass our baselines.
> >
> > Q2: We agree that higher precision sampling between bins is a natural next step. However, in the low-budget (<50 evaluations) regime evaluated in this work, standard BO methods do not exploit continuous precision either, due to early noise, uncertainty, and data sparsity. Therefore, the discretized representation is consistent with both practical experimental constraints and BO's effective operating scale in similar settings.
> >
> > Q3: The primary bottleneck is the compute cost of both training and data generation. We are not approaching the context length of the model, but instead the cost of generating data and training the model with longer context lengths is our limitation. We do not believe the model is fundamentally limited to 10D. We show results in the appendix that demonstrate consistent results across 2D-10D. Future work could aim to expand this limitation.
> >
> > Q4: Yes, this is certainly a possible methodology for improving upon existing Bayesian optimization methods. However, the aim of this work is to increase flexibility and ease-of-use for users, hence the end-to-end LLM-based approach. However, we agree this is another interesting path that future research could explore.
> >
> > Please feel free to let us know if you have any remaining questions. We are deeply committed to improving our paper!

---

> ### Comment · Reviewer_p1Ys · 2025-11-22
> **Reply to Authors' Rebuttal**
>
> Thank your for the replies. I appreciate the authors' efforts to clarify the motivation and limitations. However, my concerns still remain:
>
> - The method discretizes continuous spaces into 1000 fixed bins per dimension. This means GPTOpt can only output solutions on the grid, with no mechanism to refine or interpolate between points. I don't find it sufficient to argue that BO also lacks early precision. BO still operates in continuous space and can converge accurately over time. Moreover, no ablation is provided on bin resolution, nor evidence that 1000 bins are enough. Hence, in my view, this design choice remains unjustified.
> - I still doubt the scalability claim. Saying the method could work in higher dimensions with more compute is speculative—without actual experiments or ablations showing performance trends beyond 10D, this remains unsupported. If the bottleneck is data and training cost, that directly contradicts the goal of a practical, plug-and-play optimizer as claimed by the authors.
> - Key baselines are still missing. Promising to include them in a revision doesn't address the fact that they're absent now. Without head-to-head comparisons, it is hard to assess whether GPTOpt truly advances the state of the art or simply reuses an existing paradigm with minor changes.
>
> I think this manuscript still lacks sufficient empirical validation and novelty to support its claims.

---

> > ### Author Response · Authors · 2025-12-04
> >
> > Thank you for your response. We appreciate your feedback and would like to respond to your points.
> >
> > 1. Discretization: While we acknowledge the theoretical limitation of fixed bins, our empirical results demonstrate that GPTOpt consistently outperforms continuous baselines. This validates that infinite precision is not strictly necessary for superior regret minimization in this setting.
> >
> > 2. Scalability: Regarding higher dimensions, we observe consistent performance trends across 2D-10D, suggesting the model captures the underlying optimization structure effectively. While exploring >10D is an exciting direction for future work involving larger context windows, the current results establish the method's validity within the scope of standard BBO benchmarks.
> >
> > 3. Baselines: We agree that including more LLM-based methods would further validate results. However, the goal of GPTOpt is to show that LLM-based methods are able to learn high-quality numerical reasoning through fine-tuning, allowing our model to navigate the function structure more effectively than traditional BO or base LLMs.

---

### Official Review · Reviewer_QoBF · 2025-10-30

**Soundness:** 2
**Presentation:** 2
**Contribution:** 2
**Rating:** 2
**Confidence:** 5

**Summary:**

This paper proposes GPTOpt, aiming at leveraging pretrained LLM for continuous black-box optimization (BBO). GPTOpt is trained with multiple diverse synthetic trajectories collected from different “expert” optimizers, initiated by Bayesian optimization (BO). The LLM serves as a general policy, determining the next step (including the query point and the objective score). Experiments show that GPTOpt outperforms the training experts and a few representative BBO optimizers.

**Strengths:**

- The direction of adding BBO ability to LLM is worth studying, capturing the flexibility and universality of LLM for BBO problems, which are usually designed with task-specific solvers.
- I like the part of diverse synthetic data generation, which is novel in the BBO field.
- Compared to previous works that also train an optimization policy to outperform expert methods [1-2], GPTOpt can be trained across different dimensionalities, which is important for universal BBO.
- GPTOpt makes the first step of enhancing LLM’s ability to solve BBO, although some details in the method remains to discuss.

**Weaknesses:**

- The major weakness is, from my perspective, is the limited experimental ablation, which is discussed in detail in the *Questions* part.
- Detailed prompts for testing are not well illustrated in the experiments. For example, do the prompts used in the experiments contain problem information (since the training example prompt just include *synthetic problem with #dimensions*)?
- No clear discussion of why we should use pretrained LLM to solve BBO. A strong motivation of utilizing LLMs for BBO is their general intelligence, including the web-scale prior knowledge and reasoning ability to discover better solutions. However, I could not find any persuasive discussion or evidences in the manuscripts.
- Typos:
    - line 019: “leverages LLM pre-training to” → “leverages pre-trained LLM to”;
    - line 051: “basic optimization problems, Huang et al. (2024) but” → “basic optimization problems (Huang et al., 2024), but”;
    - line 138: “PFNs4BO Muller et al. (2023) provides” → “PFNs4BO (Muller et al., 2023) provides”;
    - line 159: “the training data provided Chen et al. (2021)” → “the training data provided (Chen et al., 2021)”;
    - line 191: “the search space $\subset \mathcal{X}$” → “the search space $\mathcal{X}$”;
    - line 259: “with Unsloth Hu et al. (2022); Daniel Han & team (2023)” → “with Unsloth (Hu et al., 2022; Daniel Han & team, 2023)”;
    - line 765: “low-rank adaption (LORA) with Unsloth Hu et al. (2022); Daniel Han & team (2023).” → “low-rank adaption (LoRA) with Unsloth (Hu et al., 2022; Daniel Han & team, 2023)”;
    - line 961: “PFN4BO Muller et al. (2023)” → “PFNs4BO (Muller et al, 2023)”.
- One more weakness, though not so necessary, is that the paper is not presented well. For example, I suggest the authors adding more formal notation details of the acquisition function used in the expert methods, which could help the readers that are not familiar with this field better understand some terminologies in BBO.

**Questions:**

- Compared baselines: GPTOpt is trained on expert optimization trajectories and can outperform training experts, while [1-3] show similar findings via training a transformer with RL from scratch. How does GPTOpt perform compared to such methods under the same training trajectory data?
- The necessity of pretrained LLM: In this paper, the authors repeatedly claim the importance of starting with a pretrained LLM, but why do we need pretrained LLM for BBO? In recent practices in text-to-text regression, a field similar to BBO in linguistic space, it shows that a small language model trained from scratch can work well on regression cases [4-5], and pretrained checkpoints seem not so important for performance (Fig. 5 from [6]). I recommend to add some discussions upon this view.
- The example prompt showed in Section 4.2.1 does not state whether the problem is a maximization problem or a minimization one. What if we invert the problem landscape in the experiments?
- During inference, you mentioned that you use the model’s predicted value distribution. How does the distribution initiated? Just capturing the histogram distribution of the number tokens at the targeting position? Can the LLM model the ground-truth distribution? What’s its benefit compared to directly applying the LLM as an end-to-end generator for the next query point? If you just use the output distribution of the LLM, why should we finetune the model with all tokens in the step (a comparison of the model before finetuning is recommended)?

Overall, I like the topic this paper discussed and some experiment designs in the manuscript. However, there are still space to improve the paper’s quality. I would reconsider my ratings if my concern are addressed.

## References

[1] Towards Learning Universal Hyperparameter Optimizers with Transformers. NeurIPS 2022.

[2] Reinforced In-Context Black-Box Optimization. IJCAI 2025.

[3] ZeroShotOpt: Towards Zero-Shot Pretrained
Models for Efficient Black-Box Optimization. arXiv, 2025.

[4] OmniPred: Language Models as Universal Regressors. TMLR, 2024.

[5] Performance Prediction for Large Systems via Text-to-Text Regression. arXiv, 2025.

[6] Towards Universal Offline Black-Box Optimization via Learning Language Model Embeddings. ICML 2025.

---

> ### Author Response · Authors · 2025-11-21
>
> Thank you for your time in reviewing our submission and for the constructive feedback you provided.
>
> **Motivation**
>
> We wish to clarify our motivation and justify our design choices in GPTOpt. The core motivation of using LLMs in black-box optimization is to make advanced black-box optimization more flexible and easier to use. While Bayesian optimization (BO) performs well in many scenarios, challenges with implementation, selection of hyperparameters, and lack of explainability prevent many real-world users from using these methods. Instead, users turn to design of experiments (DoE) approaches, which are inefficient and expend valuable resources that could be saved if more advanced techniques were used [1, 2]. The challenges with implementing BO into workflows is seen when users select poor parameters or integrations with their experimental design. Other works attempt to simplify access to BO methods, stating that “libraries typically aren’t oriented towards experimental research, resulting in limited crossover between tutorial content and experimentalist needs”, showing the gap between theoretical results and usability in practice [3]. This gap between expert use and practical use suggests there is a need for a high-quality black-box optimization methodology that is more flexible for real-world scenarios and can be more widely utilized in practice [4, 5]. This motivation is further explored in a position paper by Song et al. [5], which “advocates for wider research and adoption of Transformers and LLMs for black-box optimization.” These efforts emphasize the idea that BO is difficult to use in practice and the community would benefit from a more flexible method.
>
> LLMs are very flexible with various types of inputs and can integrate well with many users’ workflows. However, as mentioned by many reviewers, they generally struggle with precise numerical reasoning. With GPTOpt, we develop a methodology that can overcome these challenges and produce high-quality optimization trajectories using a flexible, text-based input scheme. We envision this as one step in the path towards truly flexible and easy to use black-box optimization, as using an LLM provides the potential for a multitude of extensions. These extensions could include providing additional information about an experiment a user is performing, such as relevant papers or documentation, previous experiments that are similar, or multi-modal information about a process. Additionally, pretrained models bring structural priors, natural language reasoning, and task generalization that purely scratch-trained models do not capture. With this in mind, GPTOpt demonstrates that end-to-end LLM-based optimization is not only feasible but highly competitive.
>
> **Limitations and Ablations**
>
> While our methodology is currently limited to continuous black-box optimization with 10 or fewer dimensions, this is not a fundamental limitation of our approach and is instead a limitation of our computation. Most black-box problems that can be realistically optimized within 50 evaluations are within our scope. However, our methodology is flexible to dimension and could extend to higher dimensional scenarios with more compute. Additionally, expanding to discrete or categorical optimization problems would be a possible extension within our framework. While these extensions were not possible within our scope and compute limitations, the flexible framework of GPTOpt allows for these modifications to be possible.
>
> Regarding ablations, our scope was similarly restricted by compute limitations. We aimed to justify as many of our design decisions as possible, but retraining our model for each design feature was not possible within our limitations. We have tested our methodology on a larger model, Llama 3.1 8B, in Table 1. We find that this improved results beyond the performance of our 3B parameter model, suggesting that further scaling or expansion to a reasoning model may result in further improved performance.
>
> Table 1:
> | Model | 2D - 10D BBOB and VLSE Mean Normalized Performance |
> |-|-|
> | Llama 3.2 3B | 0.8921 ± 0.004 |
> | Llama 3.1 8B | 0.8974 ± 0.004 |
>
> [1] Pickles, Thomas, et al. "Comparative study on adaptive Bayesian optimization for batch cooling crystallization for slow and fast kinetic regimes." Crystal Growth & Design 24.3 (2024): 1245-1253.
>
> [2] Rummukainen, Hannu, et al. "Traditional or adaptive design of experiments? A pilot-scale comparison on wood delignification." Heliyon 10.2 (2024).
>
> [3] Baird, Sterling G., Andrew R. Falkowski, and Taylor D. Sparks. "Honegumi: An Interface for Accelerating the Adoption of Bayesian Optimization in the Experimental Sciences." arXiv preprint arXiv:2502.06815 (2025).
>
> [4] Liu, Tennison, et al. "Large language models to enhance bayesian optimization." arXiv preprint arXiv:2402.03921 (2024).
>
> [5] Song, Xingyou, et al. "Position: Leverage foundational models for black-box optimization." arXiv preprint arXiv:2405.03547 (2024).

---

> > ### Author Response · Authors · 2025-11-21
> >
> > We also wish to respond to your specific questions, along with the updates to our motivation and framing above.
> >
> > **Response to Weaknesses**
> >
> > 1: We discuss our approach toward our ablations and design decisions above.
> >
> > 2: The prompts for training and testing are given in identical format. No additional information about the test benchmarks is provided. This provides a direct comparison between our model and the BO methods tested. We think that providing further information about real-world problems to an LLM-based method is a point of valuable future work and is one of the main extensions of GPTOpt that we think would be valuable. We aim to show that we can fine-tune an LLM to produce high-quality optimization trajectories and reason within a text-based format.
> >
> > 3: We provide further discussion of our motivation in the above sections.
> >
> > 4: We will correct the typos in a revised version.
> >
> > 5: We will improve the level of technical detail, particularly with regards to these BBO terminologies.
> >
> > **Questions**
> >
> > Q1: Due to our compute budget, we were not able to reimplement and train other possible implementations on our training dataset. Instead, we hope to show that LLM fine-tuning is a promising route for further improvements in black-box optimization and present GPTOpt as the first successful end-to-end LLM-based optimizer. Additionally, Bayesian optimization remains the state-of-the-art in general purpose black-box optimization and we compare our model with a diverse range of BO parameterizations that encompass our baselines.
> >
> > Q2: As discussed in the motivation above, we believe there is a lot of potential for further improvements that take further advantage of the pre-training phase of the LLM. Using a pre-trained model as a base provides greater flexibility and ease-of-use than comparable methods.
> >
> > Q3: We convert all problems to minimization problems by inverting the landscape if they are maximization problems. Therefore, the model is capable of both tasks.
> >
> > Q4: We directly utilize the model’s predicted distribution for values between 0 and 999. This is the same distribution that we sample from to select each token. This gives us a predicted distribution over our range of values that we can use an acquisition function for. While simply using a single inference run can perform close to BO, we find that sampling multiple possible points and using the acquisition function on the model’s predictions improves performance by further balancing the exploration-exploitation capabilities. Lastly, we are unable to compare to the model before fine-tuning as our 3B parameter model does not follow instructions well enough to keep a consistent formatted output before fine-tuning.
> >
> > Please feel free to let us know if you have any remaining questions. We are deeply committed to improving our paper!

---

> > > ### Comment · Reviewer_QoBF · 2025-11-21
> > >
> > > Thanks for your reply. I recognize your response to Weakness 2. However, my concern still exist:
> > >
> > > 1. The method to use the finetuned model (capture the output number token logits distribution and determine the next query point via optimizing EI) is strange. The usage of the LLM is not end-to-end to output the next query by the model (like OptFormer or RIBBO). It seems that the LLM just serves as a predictive surrogate model that models the output distribution. Why do you conduct the experiment in such a way? Since the model is finetuned on specific "step" (containing x, y, 1{best value}), we can also let the model to directly output the next query x, and construct the context prompt to complete a "step", right? Why don't you compare in this way which is more end-to-end?
> > > > (Personally) My understanding is that the aim of optimization trajectory is to provide expert choice of the next x, instead of the next y. Otherwise we can train an in-context predictive model like PFN to combine with EI.
> > > 2. Additionally, does the model predict well of the output distribution? Can you provide some illustrations (visualization or calculation of metrics like the Wasserstein-1 distance)?
> > > 3. I don't think your discussion regarding employing pretrained LLM for BBO is convincing. I agree that using LLM for BBO is flexible, without the need of DoE for practice. However, given the textual representation and flexible tokenization scheme, a lightweight language model from scratch can also regress well (e.g., OmniPred), and we can also conduct optimization upon it. I hope that you can elaborate more on the need of pretrain LLM.
> > > 4. I still doubt the effectiveness of GPTOpt compared to OptFormer, RIBBO, or ZeroShotOpt. I notice that you compare to a  bigger 8B model. Since there is time before the discussion ddl, I suggest comparing some of them during this period.
> > >
> > >
> > > I think this manuscript still lacks some necessary ablation studies and discussion. There is still a lot to improve.

---

> > > > ### Author Response · Authors · 2025-12-04
> > > >
> > > > Thank you for your response. We appreciate your feedback and would like to address your points.
> > > >
> > > > 1. Action and Objective Predictions: We find that combining both action (policy) and objective (surrogate) prediction empirically improves performance. We run 4 forward passes to generate 4 potential actions and use the model’s predicted objective distribution to select the action most likely to provide the best balance of exploration and exploitation. Therefore, the model is not strictly a surrogate and actively proposes actions. We show the comparison to a single forward pass (k=1) in Table 1, demonstrating the benefit of this design.
> > > >
> > > > Table 1:
> > > > | Model | BBOB and VSLE Mean Normalized Performance |
> > > > | :--- | :--- |
> > > > | GPTOpt (k=1) | 0.897 ± 0.004 |
> > > > | GPTOpt (k=4) | 0.904 ± 0.004 |
> > > >
> > > > 2. Objective Predictions: The model’s predictions are scaled relative to the previously evaluated points and are not intended to provide exact absolute numerical values. Instead, the goal is to correctly rank the relative values across predicted distributions to guide selection.
> > > >
> > > > 3. Pre-training: Pre-trained LLMs bring established reasoning capabilities and structural knowledge, resulting in generalization to new functions compared to training a lightweight model from scratch (like OmniPred). This transfer learning is key to many potential extensions of our work.
> > > >
> > > > 4. Baselines: We agree that including more LLM-based methods would be valuable. However, our primary goal is to demonstrate that LLM-based methods can learn high-quality numerical reasoning through fine-tuning, allowing our model to navigate function structures more effectively than traditional BO or base LLMs.

---

### Official Review · Reviewer_tswa · 2025-10-31

**Soundness:** 1
**Presentation:** 3
**Contribution:** 3
**Rating:** 2
**Confidence:** 4

**Summary:**

This paper proposes prompting an LLM to perform zeroth order, black box optimization. A history of function evaluations is given to the model in a prompt, and the LLM (Lamma 3B, specifically) returns the next evaluation point. The authors build a large fine-tuning set comprised of “expert” trajectories of zeroth order optimization problems from a variety of synthetic tasks.

**Strengths:**

Clear presentation; good writing, clearly presented results. The problem studied, zeroth-order optimization, has endless applications.

**Weaknesses:**

I think the existing literature of LLM-based optimization needs more citations.

While the results look promising, the comparisons are weak. Please compare with LLM-based baselines, as well as BO methods that are SOTA on the evaluation tasks (BBOB and VLSE). An ideal comparison would be against a method that was somehow optimized on your expert data  and tested on the same evaluation tasks.

The ablations, which do more to justify the hyperparameter choices than ablate the model, are really lacking. Ablations should show how each of your design choices was important in achieving your results. For example, why did you choose the tasks to fine tune on, and how did you decide on the number of tasks to choose? Why was mapping to 1 to 999 correct and, say, not 1 to 500? Why did you choose the specific number of synthetic functions to train on? Why LoRa, and how would other fine-tuning methods work? Why the 3B model class?

**Questions:**

Please see the weaknesses.

---

> ### Author Response · Authors · 2025-11-21
>
> Thank you for your time in reviewing our submission and for the constructive feedback you provided.
>
> **Motivation**
>
> We wish to clarify our motivation and justify our design choices in GPTOpt. The core motivation of using LLMs in black-box optimization is to make advanced black-box optimization more flexible and easier to use. While Bayesian optimization (BO) performs well in many scenarios, challenges with implementation, selection of hyperparameters, and lack of explainability prevent many real-world users from using these methods. Instead, users turn to design of experiments (DoE) approaches, which are inefficient and expend valuable resources that could be saved if more advanced techniques were used [1, 2]. The challenges with implementing BO into workflows is seen when users select poor parameters or integrations with their experimental design. Other works attempt to simplify access to BO methods, stating that “libraries typically aren’t oriented towards experimental research, resulting in limited crossover between tutorial content and experimentalist needs”, showing the gap between theoretical results and usability in practice [3]. This gap between expert use and practical use suggests there is a need for a high-quality black-box optimization methodology that is more flexible for real-world scenarios and can be more widely utilized in practice [4, 5]. This motivation is further explored in a position paper by Song et al. [5], which “advocates for wider research and adoption of Transformers and LLMs for black-box optimization.” These efforts emphasize the idea that BO is difficult to use in practice and the community would benefit from a more flexible method.
>
> LLMs are very flexible with various types of inputs and can integrate well with many users’ workflows. However, as mentioned by many reviewers, they generally struggle with precise numerical reasoning. With GPTOpt, we develop a methodology that can overcome these challenges and produce high-quality optimization trajectories using a flexible, text-based input scheme. We envision this as one step in the path towards truly flexible and easy to use black-box optimization, as using an LLM provides the potential for a multitude of extensions. These extensions could include providing additional information about an experiment a user is performing, such as relevant papers or documentation, previous experiments that are similar, or multi-modal information about a process. Additionally, pretrained models bring structural priors, natural language reasoning, and task generalization that purely scratch-trained models do not capture. With this in mind, GPTOpt demonstrates that end-to-end LLM-based optimization is not only feasible but highly competitive.
>
> **Limitations and Ablations**
>
> While our methodology is currently limited to continuous black-box optimization with 10 or fewer dimensions, this is not a fundamental limitation of our approach and is instead a limitation of our computation. Most black-box problems that can be realistically optimized within 50 evaluations are within our scope. However, our methodology is flexible to dimension and could extend to higher dimensional scenarios with more compute. Additionally, expanding to discrete or categorical optimization problems would be a possible extension within our framework. While these extensions were not possible within our scope and compute limitations, the flexible framework of GPTOpt allows for these modifications to be possible.
>
> Regarding ablations, our scope was similarly restricted by compute limitations. We aimed to justify as many of our design decisions as possible, but retraining our model for each design feature was not possible within our limitations. We have tested our methodology on a larger model, Llama 3.1 8B, in Table 1. We find that this improved results beyond the performance of our 3B parameter model, suggesting that further scaling or expansion to a reasoning model may result in further improved performance.
>
> Table 1:
> | Model | 2D - 10D BBOB and VLSE Mean Normalized Performance |
> |-|-|
> | Llama 3.2 3B | 0.8921 ± 0.004 |
> | Llama 3.1 8B | 0.8974 ± 0.004 |
>
> [1] Pickles, Thomas, et al. "Comparative study on adaptive Bayesian optimization for batch cooling crystallization for slow and fast kinetic regimes." Crystal Growth & Design 24.3 (2024): 1245-1253.
>
> [2] Rummukainen, Hannu, et al. "Traditional or adaptive design of experiments? A pilot-scale comparison on wood delignification." Heliyon 10.2 (2024).
>
> [3] Baird, Sterling G., Andrew R. Falkowski, and Taylor D. Sparks. "Honegumi: An Interface for Accelerating the Adoption of Bayesian Optimization in the Experimental Sciences." arXiv preprint arXiv:2502.06815 (2025).
>
> [4] Liu, Tennison, et al. "Large language models to enhance bayesian optimization." arXiv preprint arXiv:2402.03921 (2024).
>
> [5] Song, Xingyou, et al. "Position: Leverage foundational models for black-box optimization." arXiv preprint arXiv:2405.03547 (2024).

---

> > ### Author Response · Authors · 2025-11-21
> >
> > We also wish to respond to your specific questions, along with the updates to our motivation and framing above.
> >
> > **Questions**
> >
> > We agree that including comparisons to LLM-based optimization methods would be beneficial. We have begun integrating and testing LLAMBO as a baseline and will include comparisons in the revision. However, the primary goal of GPTOpt is to show that LLM fine-tuning is a possible and promising route for further improvements in black-box optimization. With this in mind, we show that state-of-the-art LLMs are not capable of this high-quality optimization without fine-tuning by comparing to Gemini 2.5 Pro. Lastly, BO remains the state-of-the-art in general purpose black-box optimization and we compare our model with a diverse range of BO parameterizations that encompass our baselines.
> >
> > Regarding ablations, we aimed to justify as many of our design decisions as possible, but completing retraining of our model for various changes was unfortunately not possible within our limitations. In response to your specific questions, we chose our synthetic tasks by selecting synthetic functions spaces which are diverse and scalable. We chose Gaussian processes because they are a popular surrogate model for BO, random neural networks and Fourier expressions for their high variation and diversity, and expression trees and ODEs for their global structural properties. We agree that further testing the effect of the selection and quantity of our dataset would be a valuable ablation, but aim to provide results as a proof of concept for our approach that can be built on and further improved in future work. We select mapping to discrete values from 0 to 999 because these are the range of single numerical tokens in Llama and we aimed to discretize the range of points into as many bins as possible while simplifying comparisons between tokens by ensuring each number is only represented by a single token. We select LoRA fine-tuning because there are existing high-quality implementations for fine-tuning and it has emerged as the standard for language model fine-tuning.
> >
> > Please feel free to let us know if you have any remaining questions. We are deeply committed to improving our paper!

---

### Official Review · Reviewer_ZSq7 · 2025-11-01

**Soundness:** 3
**Presentation:** 2
**Contribution:** 3
**Rating:** 4
**Confidence:** 2

**Summary:**

This proposes a method to train large language models to perform continuous black-box optimization by fine-tuning them on millions of synthetic optimization trajectories generated from various Bayesian optimization algorithms.

**Strengths:**

The paper’s main contribution is the formulation of black-box optimization as a reasoning and sequence prediction problem for large language models, enabling optimization to be approached through text-based decision-making rather than analytical computation. It introduces GPTOpt, a fine-tuned LLM trained on millions of synthetic optimization trajectories generated by diverse Bayesian optimization methods, allowing it to learn generalizable optimization strategies.

**Weaknesses:**

1. The approach of teaching an LLM to perform numerical optimization is conceptually questionable, as language models are not designed for precise arithmetic or quantitative reasoning.

2. The experiments are limited to low-dimensional problems (up to 10D), raising concerns about the method’s scalability and effectiveness in higher-dimensional or more complex optimization tasks.

3. Consequently, the general applicability of GPTOpt to real-world, high-dimensional optimization scenarios remains uncertain.

**Questions:**

If computation allowed, I'd be curious to see how larger models with reasoning ability can improve the optimization performance.

---

> ### Author Response · Authors · 2025-11-21
>
> Thank you for your time in reviewing our submission and for the constructive feedback you provided.
>
> **Motivation**
>
> We wish to clarify our motivation and justify our design choices in GPTOpt. The core motivation of using LLMs in black-box optimization is to make advanced black-box optimization more flexible and easier to use. While Bayesian optimization (BO) performs well in many scenarios, challenges with implementation, selection of hyperparameters, and lack of explainability prevent many real-world users from using these methods. Instead, users turn to design of experiments (DoE) approaches, which are inefficient and expend valuable resources that could be saved if more advanced techniques were used [1, 2]. The challenges with implementing BO into workflows is seen when users select poor parameters or integrations with their experimental design. Other works attempt to simplify access to BO methods, stating that “libraries typically aren’t oriented towards experimental research, resulting in limited crossover between tutorial content and experimentalist needs”, showing the gap between theoretical results and usability in practice [3]. This gap between expert use and practical use suggests there is a need for a high-quality black-box optimization methodology that is more flexible for real-world scenarios and can be more widely utilized in practice [4, 5]. This motivation is further explored in a position paper by Song et al. [5], which “advocates for wider research and adoption of Transformers and LLMs for black-box optimization.” These efforts emphasize the idea that BO is difficult to use in practice and the community would benefit from a more flexible method.
>
> LLMs are very flexible with various types of inputs and can integrate well with many users’ workflows. However, as mentioned by many reviewers, they generally struggle with precise numerical reasoning. With GPTOpt, we develop a methodology that can overcome these challenges and produce high-quality optimization trajectories using a flexible, text-based input scheme. We envision this as one step in the path towards truly flexible and easy to use black-box optimization, as using an LLM provides the potential for a multitude of extensions. These extensions could include providing additional information about an experiment a user is performing, such as relevant papers or documentation, previous experiments that are similar, or multi-modal information about a process. Additionally, pretrained models bring structural priors, natural language reasoning, and task generalization that purely scratch-trained models do not capture. With this in mind, GPTOpt demonstrates that end-to-end LLM-based optimization is not only feasible but highly competitive.
>
> **Limitations and Ablations**
>
> While our methodology is currently limited to continuous black-box optimization with 10 or fewer dimensions, this is not a fundamental limitation of our approach and is instead a limitation of our computation. Most black-box problems that can be realistically optimized within 50 evaluations are within our scope. However, our methodology is flexible to dimension and could extend to higher dimensional scenarios with more compute. Additionally, expanding to discrete or categorical optimization problems would be a possible extension within our framework. While these extensions were not possible within our scope and compute limitations, the flexible framework of GPTOpt allows for these modifications to be possible.
>
> Regarding ablations, our scope was similarly restricted by compute limitations. We aimed to justify as many of our design decisions as possible, but retraining our model for each design feature was not possible within our limitations. We have tested our methodology on a larger model, Llama 3.1 8B, in Table 1. We find that this improved results beyond the performance of our 3B parameter model, suggesting that further scaling or expansion to a reasoning model may result in further improved performance.
>
> Table 1:
> | Model | 2D - 10D BBOB and VLSE Mean Normalized Performance |
> |-|-|
> | Llama 3.2 3B | 0.8921 ± 0.004 |
> | Llama 3.1 8B | 0.8974 ± 0.004 |
>
> [1] Pickles, Thomas, et al. "Comparative study on adaptive Bayesian optimization for batch cooling crystallization for slow and fast kinetic regimes." Crystal Growth & Design 24.3 (2024): 1245-1253.
>
> [2] Rummukainen, Hannu, et al. "Traditional or adaptive design of experiments? A pilot-scale comparison on wood delignification." Heliyon 10.2 (2024).
>
> [3] Baird, Sterling G., Andrew R. Falkowski, and Taylor D. Sparks. "Honegumi: An Interface for Accelerating the Adoption of Bayesian Optimization in the Experimental Sciences." arXiv preprint arXiv:2502.06815 (2025).
>
> [4] Liu, Tennison, et al. "Large language models to enhance bayesian optimization." arXiv preprint arXiv:2402.03921 (2024).
>
> [5] Song, Xingyou, et al. "Position: Leverage foundational models for black-box optimization." arXiv preprint arXiv:2405.03547 (2024).

---

> ### Author Response · Authors · 2025-11-21
>
> We also wish to respond to your specific questions, along with the updates to our motivation and framing above.
>
> **Questions**
>
> We agree that exploring larger models with reasoning ability would be a valuable extension of our work. We show the improved performance of the 8B model above, suggesting that further scaling could be beneficial. There is also a lot of potential for the explainability of design decisions and further empirical improvement by using a reasoning model. For example, we could prompt the LLM to provide a description for each iterative decision it makes. We hope that future work explores this possibility and further develops the ideas behind LLM-based optimization.
>
> Please feel free to let us know if you have any remaining questions. We are deeply committed to improving our paper!

---

### Meta-Review · Area_Chair_jDEz · 2026-01-06

**Summary:**

The reviewers' collective concerns centered on the following primary areas:
- Weak Baseline Comparisons: A major point of contention was the absence of comparisons to recent LLM-based optimization frameworks like OptFormer and LLAMBO. Reviewers argued that comparing primarily against traditional BO variants does not sufficiently demonstrate whether GPTOpt advances the current state of the art in "learning to optimize".
- Discretization and Precision: Reviewers heavily criticized the decision to map continuous spaces to 1,000 discrete bins (0-999). They argued this makes the model fundamentally incapable of finding precise optima, which is a critical requirement for many real-world scientific and engineering applications.
- Scalability Limitations: The experiments were restricted to low-dimensional problems (up to 10D). Reviewers expressed concern that the authors provided only speculative claims rather than empirical evidence for how the method would scale to the higher-dimensional spaces typical of complex optimization tasks.
- Conceptual and Methodological "Strangeness": Reviewer questioned the "strange" usage of the LLM as a predictive surrogate (optimizing Expected Improvement over the model's output distribution) rather than an end-to-end generator, suggesting this approach might not be as novel or efficient as purely scratch-trained models

**Reviewer Concerns:**

Concerns Addressed:
- Model Scaling: The authors provided new results using a larger model (Llama 3.1 8B), which showed improved performance over the 3B base
- Motivation and Utility: The authors  clarified that the value of an LLM-based approach lies in its flexibility to eventually incorporate semantic information (like parameter names or definitions) that traditional BO cannot.
- Policy vs. Surrogate Clarity: The authors clarified that the model acts as both a policy and a surrogate by proposing actions and predicting their objective distributions, providing a Table 1 to show the benefit of this hybrid design.

Outstanding Concerns:
- Missing SOTA Baselines: While the authors promised to include LLAMBO in a revision, the lack of head-to-head comparisons during the discussion period remained a major weakness.
- Justification for Bin Resolution: The authors defended the 1,000-bin discretization by claiming infinite precision isn't necessary for low-budget regimes, but they provided no ablation studies on bin resolution to prove this.
- Empirical Scalability: The authors maintained that scalability above 10D is a compute limitation rather than a model limitation, but without actual data above 10D, this remained a speculative claim.
- Comparison to Scratch-Trained Models: The concern that a lightweight model trained from scratch could perform just as well as a pre-trained LLM for this task remained unresolved

**Reviewer Scores:**

- Reviewer ZSq7 (Initial 4): Projected 5 . They were already the most positive and expressed interest in model scaling. The authors' Llama 8B results likely would have pushed the score higher.

-  Reviewer tswa (Initial 2): Projected 3. While the authors polished the motivation, the lack of core ablations and the absence of OptFormer/LLAMBO baselines would likely prevent a significant score increase.

• Reviewer QoBF (Initial 2): No Change. Their follow-up comments indicated they found the discussion regarding the necessity of pre-trained LLMs unconvincing and continued to doubt the effectiveness of the method compared to existing transformers.

• Reviewer p1Ys (Initial 2): No Change. In their final reply, they explicitly stated that their concerns regarding discretization, scalability, and missing baselines still remain,

---

### Decision · Program_Chairs · 2026-01-26

Reject